New record of Frilled shark Chlamydoselachus anguineus Garman, 1884 (Chondrichthyes: Hexanchiformes) in the South Pacific Ocean

Indurain Maria J. 1 2
Mancilla Angel 1 2
Brizuela Leandro 1 2
Vargas-Caro Carolina 1 2
http://orcid.org/0000-0002-0816-6406 Bustamante Carlos 1 2 carlos.bustamante@uantof.cl
1 Programa de Conservación de Tiburones, Universidad de Antofagasta , Antofagasta , Chile
2 CHALLWA, Laboratorio de Biología Pesquera, Instituto de Ciencias Naturales Alexander von Humboldt , Universidad de Antofagasta, Antofagasta , Chile
Doğdu Servet
Electronic publication date: 2025 Feb 7
Publication date: 2025
Volume: 13
Electronic Location ID: e18992
Received 2024 Oct 18; Accepted 2025 Jan 23
Copyright: © 2025 Indurain et al.
Copyright year: 2025
Copyright holder: Indurain et al.
License: This is an open access article distributed under the terms of the Creative Commons Attribution License, which permits unrestricted use, distribution, reproduction and adaptation in any medium and for any purpose provided that it is properly attributed. For attribution, the original author(s), title, publication source (PeerJ) and either DOI or URL of the article must be cited.
License URL: https://creativecommons.org/licenses/by/4.0/

Keywords: Chlamydoselachus, Chile, Body morphometrics, Museum collections

Funding: Agencia Nacional de Investigación y Desarrollo de Chile (ANID) 11220358 This study was funded by Agencia Nacional de Investigación y Desarrollo de Chile (ANID, No. 11220358). The funders had no role in study design, data collection and analysis, decision to publish, or preparation of the manuscript.

==============================
Frilled sharks are rarely observed and limited information is available regarding their life history. The global number of records for these sharks does not exceed 40, with most sightings reported in the Western Pacific Ocean. In 1976, one specimen was recorded for the first time in the South East Pacific Ocean; however, this study provides new evidence for the presence of Chlamydoselachus anguineus in Chilean waters, extending its distribution range 850 km south of the previous record. The morphological features of all specimens from Chile are discussed, supplementing previous observations and adding to the existing knowledge on this species in the Pacific Ocean.

Introduction

The frilled shark Chlamydoselachus anguineus Garman, 1884 is one of two deep-water extant sharks in the family Chlamydoselachidae. This shark is notable for its elongated body and the presence of six gill slits on the side of the head, absence of spines on the dorsal fin, which is located on the back of the body, and the presence of tricuspid teeth (Ebert, Fowler & Compagno, 2021). In contrast to C. africana (Ebert & Compagno, 2009) which is restricted to southern Africa, C. anguineus has a wide-ranging but sporadic distribution in the Atlantic and Pacific Oceans. This species usually inhabits outer continental and insular shelves and continental slopes at depths of 120–1,500 m (Shirai, 1996; Ebert, Fowler & Compagno, 2021). Although rarely observed, frilled sharks have been documented as bycatch in trawl and bottom longline fisheries throughout its range (Smart, Paul & Fowler, 2016). Frilled sharks are considered living fossils, and its life history and biology remain poorly documented despite the early interest in this species stemmed from its affinity with the cladodonts (Garman, 1885; Gill, 1884). The International Union for Conservation of Nature (IUCN) assessed the species as having a conservation status of “Least Concern”. Currently, there is no information on the population size, structure, or trends of the species. Global reports on frilled sharks are limited, with no more than 40 specimens documented since its original description (Ebert & Compagno, 2009), suggesting its relatively scarce presence across oceans. Although most observations were from the west coast of the Pacific Ocean, two specimens were reported from the east coast of the Pacific Ocean. One female was caught off California in 1948 (Noble, 1948) and another female was caught off central Chile in 1977 (Morillas, 1977). In this study, we documented a third specimen of C. anguineus in the East Pacific and the first male frilled shark found in Chilean waters. Additionally, the body morphometrics of both Chilean specimens were analyzed, supplementing the extant knowledge of the species in the South East Pacific Ocean.

Materials AND methods

In February 2015, a male frilled shark was caught during trawling by a fishing vessel off Cucao, Chiloé Island (Fig. 1). The specimen was obtained from a depth of 500 m, preserved in formalin, and stored in a private collection until 2023. A review of the diagnostic characteristics was performed following the methods described by Compagno (1984) and Ebert, Fowler & Compagno (2021). Sex was determined based on the presence of claspers in males, and sexual maturity was assessed based on clasper calcification and development of internal sexual organs (Tanaka et al., 1990). In addition, a set of 48 morphometric measurements was recorded as species descriptors, following the recommendations of Ebert & Compagno (2009) for comparative studies of the genus. Additionally, the morphometric set was obtained from the Morillas specimen which was deposited at the National Museum of Natural History (MNHN-CH, Catalogue Number P-5815).

Figure 1 Map of locations of C. anguineus in Chilean waters.

The geographical locations where specimens of C. anguineus was landed in 1976 (Santa Maria Is.) and 2015 (Cucao).

Examined material: Chlamydoselachus anguineus Garman 1884.

MNHN-CH-5815: Female, 140.5 cm total length (LT). Isla Santa María (37°03′S, 73°31′W), Concepción, Chile. Caught by longline at 550 m depth on 8 April 1976. Deposited at the Museo Nacional de Historia Natural de Chile (MNHN).

CHALLWA-UA-024: Male, 123.5 cm LT. Cucao (42°37′S, 74°07′W), Chiloé Island, Chile. Caught by trawling at 500 m depth on 2 February 2015. Deposited at the Colección Ictiológica de la Universidad de Antofagasta (CHALLWA-UA).

This study did not involve animal experimentation or harm. Specimens were obtained as bycatch from vessels that target bony fish and sharks for trade according to the Chilean Law. All work was carried out with permission from the Fisheries Undersecretariat (FIPA2021-24).

Results

The shark caught off Cucao was donated by fishermen for the study; however, the specimen deteriorated over time due to poor preservation. Morphological data were retrieved with extreme care, as mishandling could compromise the integrity of the sample. Based on the external morphology (Figs. 2A–2C), the specimen was identified as C. anguineus following the diagnostic features described by Tanaka et al. (1990), Compagno (1984), and Ebert & Compagno (2009). The colouration of the specimen was uniformly dark brown; however, it was adversely affected by the use of formalin, resulting in a light brown colour after preservation.

Figure 2 Examined specimens of Chlamydoselachus anguineus from Chilean waters.

(A) Lateral view of the body, (B) head and (C) jaws of male specimen (CHALLWA-UA-024) of C. anguineus, in life. (D) Lateral view of head and (E) jaws of female specimen (MNHN-CH-5815) of C. anguineus, in preservation. (F) Vental view of testis of male specimen (CHALLWA-UA-024), in life.

The samples from Chilean waters were characterised by an elongated eel-like body with six gill slits, a terminal mouth with narrow tricuspid teeth in the jaws, a single dorsal fin, small, lobe-like, originating far back on body, anal fin larger than dorsal fin, pectoral fins small, paddle-shaped, caudal fin with a weak ventral lobe, and no subterminal notch (Fig. 2). Additionally, the identity of the Morillas sample was compared with information from the new specimen and recent studies. No morphological differences were found between the two samples from Chilean waters. The morphometric set of both Chilean specimens is presented in Table 1, which supplements previous observations of the species. Molecular data remained undocumented for the samples, as no DNA was obtained from the tissue owing to formalin-related issues. Although the maturity of the female specimen was not recorded, the male was assessed as mature based on the calcification of the claspers and testis development (Fig. 2F). The female specimen is currently deposited in the Museum of Natural History of Chile and lacks internal organs, which precludes our ability to ascertain its state of sexual maturity.

Table 1 Measurements of the morphometric characteristics of C. anguineus from Chilean waters.

Measurements are expressed as a percentage of total length. Accession numbers for each specimen are indicated in each column, respectively.

Morphometric character	Male
(CHALLWA-UA-024)	Female
(MNHN-CH-5815)	
Total length	123.5 cm	140.5 cm	
Precaudal length	80.89	74.80	
Pre-narial length	1.33	1.63	
Pre-oral length	0.65	0.81	
Pre-orbital length	3.17	4.07	
Pre-spiracle length	5.28	6.50	
Pre-gill length	10.41	11.38	
Head length	19.51	18.29	
Pre-pectoral length	18.70	16.26	
Pre-pelvic length	52.44	51.38	
Snout-Vent length	26.67	56.75	
Vent-Caudal fin length	33.90	56.91	
Pre-anal fin length	65.04	51.06	
Pre-dorsal fin length	64.07	65.28	
Dorsal-Caudal length	3.98	4.07	
Pectoral-Pelvic length	32.52	31.71	
Pre-anal length	3.50	3.25	
Anal-Caudal length	1.40	3.58	
Eye length	1.51	1.38	
Eye height	0.41	0.81	
Interorbital width	4.96	7.72	
Nostril width	0.65	0.24	
Internarial width	3.33	3.66	
Anterior nasal flap	0.77	0.24	
Mouth width	6.33	8.54	
Mouth length	0.98	0.89	
1st gill opening length	6.27	4.07	
2nd gill opening length	5.04	4.55	
3rd gill opening length	4.64	5.28	
4th gill opening length	4.70	4.88	
5th gill opening length	4.08	4.39	
6th gill opening length	3.83	3.90	
Head height	5.53	4.88	
Had width	6.93	8.54	
Trunk height	7.69	9.76	
Trunk width	6.23	12.20	
Caudal peduncle height	5.69	4.80	
Caudal peduncle width	1.27	1.87	
Pectoral fin length	4.89	4.88	
Pectoral fin anterior margin	9.45	9.51	
Pectoral fin base	4.93	4.88	
Pectoral fin height	5.87	5.77	
Pectoral fin inner margin	5.48	5.69	
Pectoral fin posterior margin	5.33	5.45	
Pelvic fin length	11.58	7.56	
Pelvic fin anterior margin	8.83	9.92	
Tooth count (upper jaw)	13-13	12-12	
Tooth count (lower jaw)	12-1-12	12-1-12	

Discussion

New evidence of C. anguineus confirms its presence in Chilean waters, extending its distribution range to 850 km south of the previous record. Body descriptions are presented for specimens from Chile, which supplement previous observations of the species. Although the body shape and size variation were within the range described for C. africana and C. anguineus (Ebert & Compagno, 2009), the head length of Chilean specimens appeared to be larger than those from Japan (18–19% LT vs. 15–16% LT), and the pelvic fins were smaller (7.6–11.6% LT vs. 11.6–15.4% LT). According to Ebert, Fowler & Compagno (2021), the head length appears to be one of the diagnostic characteristics used to separate the two species of Chlamydoselachus (17.3–17.9% LT for C. africana vs. 13.1–16.2% LT for C. anguineus). However, pelvic fins may exhibit sexual dimorphism for the genus, and this trait may be larger in males than in females (Ebert & Compagno, 2009).

Our sample size was remarkably small and may have led to a bias in morphometric interpretation as a result of intrinsic factors, such as sexual dimorphism, and extrinsic factors, including deformation caused by preservation. However, we reported the presence of a mature male in the East Pacific, with only eight mature males documented worldwide (Ebert & Compagno, 2009).

The presence of C. anguineus has been confirmed in Chilean waters, and its distribution range in the South East Pacific Ocean has been updated, with specimens inhabiting waters off Concepción (37°S) and Chiloé Island (42°S). This record may promote an integrated Ocean basin assessment that may include specimens held in museums and private collections to supplement our knowledge of this elusive shark.

The authors would like to acknowledge R. Vega and A. Augsburger, who donated the specimen for study. Also, special thanks to the staff of “Programa de Conservación de Tiburones”.

Additional Information and Declarations

Competing Interests

The authors declare that they have no competing interests.

Author Contributions

Maria J. Indurain performed the experiments, analyzed the data, authored or reviewed drafts of the article, and approved the final draft.

Angel Mancilla performed the experiments, prepared figures and/or tables, and approved the final draft.

Leandro Brizuela performed the experiments, prepared figures and/or tables, and approved the final draft.

Carolina Vargas-Caro conceived and designed the experiments, analyzed the data, authored or reviewed drafts of the article, and approved the final draft.

Carlos Bustamante conceived and designed the experiments, analyzed the data, authored or reviewed drafts of the article, and approved the final draft.

Field Study Permissions

The following information was supplied relating to field study approvals (i.e., approving body and any reference numbers):

This study did not involve animal experimentation or harm. Specimens were obtained as bycatch from vessels that target bony fish and sharks for trade according to the Chilean Law.

All work was carried out with permission from the Fisheries Undersecretariat (FIPA2021-24).

Data Availability

The following information was supplied regarding data availability:

Specimens used for this study are publicly available at Museo de Historia Natural de Chile (Specimen MNHN-CH 5815) and Colección Ictiológica Universidad de Antofagasta (Specimen CHALLWA-UA-024).

The raw measurements are available in Table 1 and are used to compare against published morphometric data.

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
