# Peer review of "New record of Frilled shark Chlamydoselachus anguineus Garman, 1884 (Chondrichthyes: Hexanchiformes) in the South Pacific Ocean"

_PeerJ, doi:10.7717/peerj.18992_

## Round 0.1 · original submission · Major Revisions

Dear Author,

Thanks for your submission to PeerJ. Your MS can be accepted after major revisions. Please check for all reviewers' suggestions.

Best Regards Servet

·

Basic reporting

The language is clear and professional.
There is potential to site additional literature and expand on the results and discussion.
The structure of a hypothesis, prediction, result appears outside the scope of this article as written.

It has the potential to describe the first male specimen of Chlamydoselachus anguineus recovered from this region of the Pacific Ocean.

Experimental design

This article potentially most aligns with a "case report" concept among the Aims and Scope of the journal. There is no specific research question, I assume because this is a short report. Add some language to the introduction about the type of paper you are hoping to present so that it clearly aligns with Aims and Scope of the journal.

The title implies a substantial forthcoming description of morphology. There can be much more description as opposed to implied morphology and a table of morphometrics.

The abstract catches on the significance of finding another specimen (at all) and in the waters they did. This has a great potential for more discussion and impact. Please tell us more about how this matters.

The figures are clear and in focus. They will benefit from labels to point out features relevant to taxonomic placement.

Validity of the findings

"the specimen was identified as C. anguineus after recent recommendations on its taxonomic status:" and "Although the maturity of the female specimen was not recorded, the male was assessed as mature based on the calcification of claspers and testis development (Figure 1E)."
The above sentences are from the results and it conflates new finding and prior research. The format of the results section, I would expect, to be a new description of the specimen in the absence of comparison to other author's results. It skips ahead to the assumption that all readers will have a known diagnosis and description at the ready and thus, undermines the importance of the new specimen.

The discussion jumps from geographic range to morphometric range data without segue - each of those data warrant further discussion.

"...factors, such as sexual dimorphism, and extrinsic factors, including deformation caused by preservation."

The above sentence is from the discussion and points out a potential flaw with the paper as written and presented. The alleviate the concern, each topic should be discussed.

Anecdotally, absolute head length is not likely to be too deformed in an extant specimen, even if there has been some amount of dehydration. Areas of postcranial skeletal material would be more likely, and given that the caudal fin can be about 50% of the shark's total body length, I can see where relative measurements to total length could be concern for bias in dried skeletal material. I'm not worried about wet specimens as much though. Tell us more about how each specimen you observed as preserved and consider also other taxa you might have looked at for comparison. As a paleontologist, I understand what a low sample size can hide, however, it's what you have... check out some relevant literature from the fossil record which helps to describe and then deconstruct the deformation for more accurate reconstructions. What other extrinsic factors are you worried about?

Additional comments

There are some questions that should be asked and answered directly in the paper to help distinguish this paper as more than a published specimen ID, even though this is a potential "case report" - I saw potential because I think you need to review the guidelines and see if you can rearrange the arguments to fully meet the scope of the journal.

For inclusion in the results.
1) What are the similarities and differences between the two specimens?
2) How "could you" assess the maturity of the female (right now you say you don't, but you don't say why)?
3) What about the skeletal morphology?

For inclusion in the discussion
4) What does the literature already describe in terms of male and female specimens?
5) Where else are specimens found and from those areas, how many have recovered both a male and a female?
6) The pacific ocean is vast, so include a map of localities then ask, what does this tell you about behavior, overlap with other taxa, comparisons

Note: Even if there isn't much published about potential dimorphism and regional correlation, you have the chance to add to the discussion now. There are at least 200 specimens listed in FishBase. Also, museums have some radiographic images to compare with - see example: http://n2t.net/ark:/65665/m3a3185ed2-002b-481e-8e44-c590fc028a6b which you can search for more here - https://collections.nmnh.si.edu/search/fishes/?v=g0#new-search

There could be something...

Reviewer 2 ·

Basic reporting

No comment

Experimental design

No comment

Validity of the findings

No comment

Additional comments

Review of “Morphology of the frilled shark Chlamydoselachus anguineus (Chondrichthyes: Hexanchiformes) in the South Pacific Ocean”
General Comments:
Overall: This is an important record to add for our knowledge of the frilled shark. I do think an ichthyology journal would have been better for this but looks like they (JFB) are either being lazy or struggling to find reviewers. Please make some minor edits and additions.
Line 37: It would be better to rephrase as “Frilled sharks are considered living fossils”
Line 78: While I agree the Discussion should be short, I think a little more can be said about the biology of the frilled shark. Are its populations scattered? Possible migration?? Raise some additional future directions for researchers.

·

Basic reporting

Dear Authors,

You can find my comments as sticky notes on the pdf version.

Best wishes

Experimental design

The method is sufficient.

Validity of the findings

The Discussion section is not sufficient. This section is should be strengthened

Additional comments

After the corrections I suggest, I may consider it appropriate to publish the paper.

Reviewer 4 ·

Basic reporting

All my opinions and suggestions are in section 4.

Experimental design

All my opinions and suggestions are in section 4.

Validity of the findings

All my opinions and suggestions are in section 4.

Additional comments

In this study, the authors determined the morphological characteristics of the second specimen of the Chlamydoselachus anguineus species from the Chlamydoselachidae family, which is considered a living fossil, sampled in Chile. The current study also revealed that Chlamydoselachus anguineus expanded its distribution in Chilean marine waters by 850 km south compared to its previous record. I would like to make a few suggestions to the authors.

-The IUCN status of the species, its general biology (growth, feeding and reproductive characteristics) and general morphological characteristics should be stated in the Introduction.

-A sampling map including the past records of the species in the Pacific Ocean should be added to the Materials and Methods section. In this way, both the exact sampling point of the species and how it expanded its distribution will be visualized.

-The coloration of the species should be mentioned in the Results section and tooth count, cranial measurements and, if possible, vertebra count, radial count of fins and spiral valve count should be performed, where regional differences can be observed, by referring to Ebert et al (2009).

-The authors have compared the diagnostic characters of this record with Morillas (1977), but I strongly recommend that they make a comparison with C. africana at the genus level by using Ebert et al (2009) and adding an additional column to Table 1, as this will increase the comprehensiveness of the article.

-Although the authors have determined 48 morphological characters, the Discussion section of the article is very weak. A more comprehensive discussion should be made with Ebert et al (2009), especially including the genus level, to reveal the similarities or differences between morphometric characters and to discuss the microevolutionary processes that may cause this. Because morphometric characters can be affected by both temporal or spatial differences within species and can vary between species.

Therefore, I strongly recommend that the authors improve the article by taking into account the issues I have mentioned above.

---

## Round 0.2 · accepted · Accept

Dear Author,

Thanks for your submission to PeerJ. Thank you for the necessary corrections. I think the publication is at a sufficient level for a "new record" publication. Congratulations, your article has been accepted.

Best Regards
Servet